# Release of Ag/ZnO Nanomaterials and Associated Risks of a Novel Water Sterilization Technology

**Chengfang Pang [1]**, **Aiga Mackevica [1,2]**, **Jingjing Tian [3,4]**, **Hongqing Feng [3]**, **Zhou Li [3,5,*]** **and Anders Baun [1,*]**

[1]  Department of Environmental Engineering, Technical University of Denmark, 2800-Kgs. Lyngby, Denmark; chengfang.pang@yahoo.com (C.P.); aiga.mackevica@univie.ac.at (A.M.)

[2]  Centre for Microbiology and Environmental Systems Science, University of Vienna, Althanstraße 14, UZA 2, Room 2C475, A-1090 Vienna, Austria

[3]  CAS Center for Excellence in Nanoscience, Beijing Key Laboratory of Micro-Nano Energy and Sensor, Beijing Institute of Nanoenergy and Nanosystems, Chinese Academy of Sciences, Beijing 100083, China; tianjingjing@binn.cas.cn (J.T.); fenghongqing@binn.cas.cn (H.F.)

[4]  Central Laboratory, Peking Union Medical College Hospital, Peking Union Medical College and Chinese Academy of Medical Sciences, Beijing 100730, China

[5]  Center on Nanoenergy Research, School of Physical Science and Technology, Guangxi University, Nanning 530004, China

*  Correspondence: zli@binn.cas.cn (Z.L.); abau@env.dtu.dk (A.B.); Tel.: +86-010-8285-4762 (Z.L.); +45-4-525-1567 (A.B.)

**Abstract:** For water sterilization, a highly effective system utilizing electrophoresis and the antimicrobial properties of Ag/ZnO nanomaterials has been developed. However, the key component of this system, a sterilization carbon cloth containing Ag/ZnO nanomaterials, has not been evaluated with respect to the potential environmental and human health risks associated with the nanomaterials released. In this paper, a recirculation flow system and methodology were developed to study the release of Ag and ZnO during water treatment. Our study showed that the released silver nanoparticles and dissolved Ag from the carbon cloth were 50 µg/L and 143 µg/L in the United States Environmental Protection Agency (EPA) medium, respectively. The release of dissolved Zn in the EPA medium was 33 µg /L. The results indicate that the release of dissolved and nanoparticulate silver from the sterilization carbon cloth exceeded acceptable risk levels in the aquatic environment. However, if the sterilization carbon cloth was pre-washed two days prior to use, the concentration of Ag was below the drinking water limit of 0.1 mg/L. Our study provides important exposure data for a novel water sanitation technology for real-world application in waste water and drinking water treatment, and aid in assuring its safe use.

**Keywords:** nanoparticles; advanced materials; risk assessment; sustainable nanotechnologies; water treatment

---

## 1. Introduction

Water scarcity is a global problem, especially in developing countries. Poor water quality and unsustainable supply limits national economic development and can lead to adverse health and economic impacts. Nanotechnology is a key emerging technology with significant potential for innovation in water treatment [1] (e.g., using advanced materials like nanostructured photocatalysts with surface chemistries, band-edge energies and bandgaps that enable selective binding and degradation of targeted contaminants using sunlight [2,3]); using nanostructured carbon-based materials with high electronic conductivity and hierarchical porous structure as electrodes for

electrosorption/capacitive deionization to enhance desalination performance [4,5]; engineering the morphology and surface area of electrodes through the use of nanotube arrays or three-dimensional macroporous structures to improve kinetics and mass transfer in electrochemical oxidation [6–9]; functionalizing the surface of nanomaterials by organic ligands for the efficient detection and adsorption of organic or inorganic materials from contaminated water [10–20], and controlling the size of magnetic nanoparticles to enhance superparamagnetism for low-energy separation and recovery with magnets [21]. The introduction of such advanced materials in water treatment requires an assessment of the potential environmental and human health risks of these materials. This is, however, very challenging as new scientific insights at the interface of biological systems and nanomaterials are emerging alongside the development of technology [22]. Thus, sufficient and reliable data for the risk assessment of nanomaterials are often lacking [23–26]. During the life cycle of nano-enabled products, interaction of the nanomaterials with the environment occurs and this significantly affects the release and exposure to nanomaterials. However, the underlying mechanisms for the release of nanomaterials and their interactions at the nano–bio interface are far from being completely understood [27–29]. Consequently, studies addressing the release of nanomaterials during their use are urgently needed.

Silver nanoparticles (AgNPs) are one of the most widely used nanomaterials in consumer products due to their antimicrobial and antifungal properties [30,31]. AgNPs are, for example, used in wound dressings and medical textiles for topical and prophylactic antibacterial treatments, and antimicrobial air filters to prevent bioaerosols accumulating in ventilation, heating, and air-conditioning systems [30]. The antibacterial action of AgNPs has been linked to silver's ability to disrupt the function of proteins and DNA replication as well as damage bacterial cell membranes [32]. The release of nanomaterials into the environment from consumer products can happen at any point of the product life cycle, but most often, environmental release happens during use, disposal, or weathering of products [33]. However, only a limited number of studies have investigated the release of nanomaterials under use conditions, most probably due to time and cost constraints for these kinds of experiments. Therefore, it is necessary to find simple and effective methods that allow for experimental testing of nanomaterial releases during the use phase of the consumer products [34].

In this paper, we present an evaluation of the nanomaterial released from a self-powered and high-efficient water sterilization system based on several advanced materials [35]. The technology includes a wave-driven triboelectric nanogenerator (TENG) and two nanobrush electrodes made of AgNPs integrated in ZnO-nanowires (ZnO-NW). The main functional component of the system is the carbon cloth containing Ag/ZnO nanomaterials. The sterilization efficiency of this system has been shown to be high for various microbes including those in natural river water. A reduction in the colony forming units (CFU) from $10^6$ mL$^{-1}$ to 0 within 0.5 min of electrical field (EF) treatment has been documented for Gram-negative bacteria [35]. In this study, it was also shown that the sterilization ability was sustained for at least 20 min after withdrawing the EF. The sterilization efficiency was based on the synergetic impact of electricity and the antimicrobial properties of the Ag/ZnO nanomaterial (i.e., intracellular generation of reactive oxygen species (ROS)), in addition to electroporation during EF treatment [35].

Even though this new sterilization system provides a promising low-cost and high efficiency solution, potential nanomaterial released during its use could prevent an evaluated widespread application. The carbon cloth containing Ag/ZnO nanomaterials, which is the core of the system, could release AgNPs or ZnO-NW to the treated water and potentially impact humans and the environment. The aim of this study was therefore to develop a simple methodology to study the release of nanomaterials from advanced materials used in water treatment and to apply it to the sterilization carbon cloth containing Ag/ZnO nanomaterials. The findings of this study can help assess risks associated with water sterilization systems and similar applications.

## 2. Materials and Methods

### 2.1. Synthesis of Sterilization Materials

The synthesis of sterilization Ag/ZnO nanomaterials was conducted as described in our previous study [35]. A woven textile material made of carbon fiber (10 μm in diameter) was used as the substrate. The carbon cloth was cleaned with acetone and ethanol three times first, and then treated with oxygen plasma for 5 min before nanomaterial growth. After plasma treatment, a ZnO-NW seed solution was added to the carbon cloth to initiate the ZnO nanowire growth. The ZnO nanowires were grown in an aqueous solution at 95 °C for 4 h by a wet-chemical method. After that, the ZnO nanowires samples were immersed in a 100 mM AgNO$_3$ ethanol–water solution for 24 h in darkness, then rinsed with ethanol to remove unconjugated Ag ions. Finally, both sides of the composite sample were exposed under UV (370 nm) irradiation for 0.5 h to reduce the Ag ions to Ag NPs in situ on the ZnO nanowire. The nanostructure of the cloth and resulting Ag/ZnO product was analyzed by scanning electron microscope (SEM) (FEI, Quanta FEG) and transmission electron microscopy (TEM) (FEI, Tecnai G2). Ag and ZnO were analyzed by energy dispersive x-ray (EDX) analysis (Oxford Instruments). The carbon cloth (CC), carbon cloth containing ZnO-NW (ZnO-CC), and carbon cloth containing AgNPs and ZnO-NW (Ag/ZnO-CC) were weighed to determine their total weight, and their Ag and/or Zn content was analyzed by inductively coupled plasma mass spectrometry (ICP-MS) (7700x, Agilent Technologies, Santa Clara, United States). After the five days release test described below, the changes in the total weights and metal contents of the cloths were also determined.

### 2.2. Methodology for Assessing Release of Nanomaterials and Dissolved Metals

A recirculation flow system was built to analyze the release of Ag/ZnO nanomaterials from carbon cloth during water treatment using EPA hard water (NaHCO$_3$: 192 mg/L, CaSO$_4$: 120 mg/L, MgSO$_4$: 120 mg/L, KCl: 8 mg/L) (EPA, 2012) and deionized, membrane filtered water (MilliQ water) as the test medium (Figure S1). In each experiment, 300 mL of water was pumped through the sterilization carbon cloth containing Ag/ZnO nanomaterials using a peristaltic pump (the sterilization carbon cloth with a 25 mm diameter was mounted in a metal filter holder) at a flow rate of 100 mL/h (Figure S1). The water was pumped continuously through the filter and every 24 h, the liquid was collected in glass bottles and replaced with clean EPA hard water or deionized water to repeat the recirculation. The total running time was 120 h (5 days). On day 6, the system was cleaned with MilliQ water (24 h recirculation) and 10% HNO$_3$ in MilliQ water (additional 24 h recirculation). Similar experiments were conducted for reference samples that included media, carbon cloth, and carbon cloth with ZnO-NW (Figure 1).

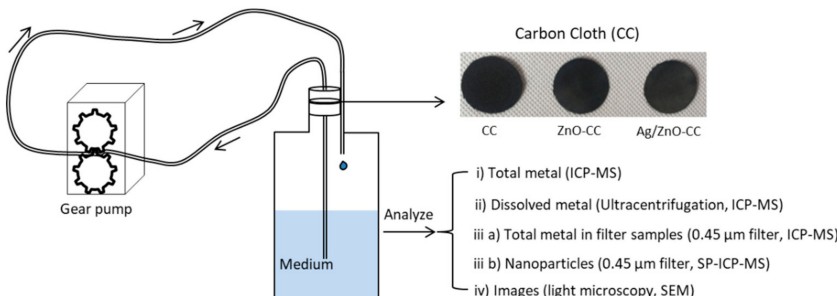

**Figure 1.** A recirculation flow system for analyzing metals released from the carbon cloth containing Ag/ZnO nanomaterials in MilliQ water and the EPA medium.

### 2.3. Characterization of Released Nanomaterials and Dissolved Metals

Water samples were collected every 24 h for five days and separated into five sub-samples for further analyses: (i) the total metal (Ag and Zn) released was analyzed by ICP-MS; (ii) releases of

dissolved Ag or Zn were analyzed in the supernatant of samples after ultracentrifugation (30,000 rpm, 30 min) by ICP-MS; (iii) collected samples were filtered through 0.45 µm filter paper (Durapore® membrane filters, Merck Millipore Ltd. Massachusetts, United States) and divided in two subsamples (iiia and iiib) where the iiia samples were analyzed by ICP-MS to quantify the total metal content in filtered samples, and the iiib samples were analyzed by single particle inductively coupled plasma mass spectrometry (SP-ICP-MS) (Perkin Elmer, NexION 350D) to quantify the release of nanomaterials; and (iv) the released fragments were analyzed by light microscopy and SEM (Figure 1). After five days of metal release from the sterilization carbon cloth containing Ag/ZnO nanomaterials, the samples of CC, ZnO-CC, and Ag/ZnO-CC were collected for Ag/ZnO mass balance analysis.

Prior to ICP-MS analysis, all samples for the determination of total metal content were incubated for 1 h at 70 °C with 1:1 $HNO_3$. External calibration was performed by the analysis of a blank and five solutions of dissolved Ag or Zn in 1% $HNO_3$ ranging from 0 to 100 µg/L. The $^{107}$Ag and $^{66}$Zn intensity for each solution was then averaged from the entire length of the analysis. Reported values were the average result of five measurements (Figure 2). SP-ICP-MS measurements were conducted for samples filtered through a 0.45 µm filter, and no additional sample preparation was performed, apart from dilution with MilliQ water if necessary. Samples were run using 100 µs dwell time for 100 s per sample. Particle size was calculated based on the dissolved metal calibration curve, and prepared in the same matrix as the experimental samples. Transport efficiency was determined based on measurements of 60 nm Au nanomaterials (Perkin Elmer, Massachusetts, Unite States). Data processing was done using Syngistix software (v.2.1, Perkin Elmer). The limit of detection (LOD) size was 23 nm for Ag and 40 nm for ZnO, assuming the spherical shape of the particles.

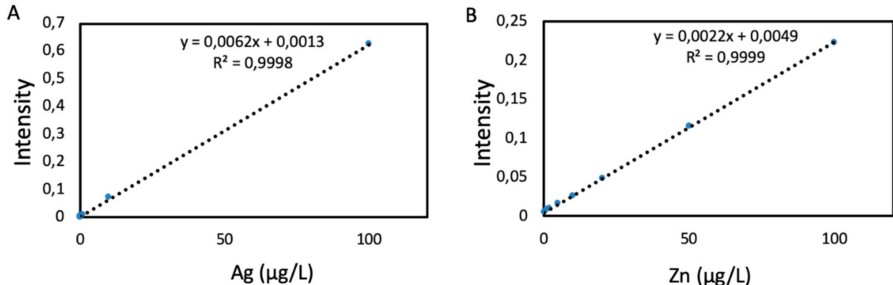

**Figure 2.** Calibration curve of Ag (**A**) and Zn (**B**) for the analysis of total metal content by inductively coupled plasma mass spectrometry (ICP-MS).

### 2.4. Statistical Analysis

A one-way analysis of variance (one-way ANOVA) with Tukey's post-hoc analysis was used to compare the statistical significance in the differences between the treatment groups. The *p*-value of 0.05 was used in the statistical tests. Data were displayed as mean (±SD) and analyzed using Origin 8 software (OriginLab Corporation, Northampton, MA, USA).

## 3. Results and Discussion

### 3.1. Characterization of the Sterilization Carbon Cloth

The TEM and SEM analysis showed that the ZnO-NW grew perpendicular to the CC (Figure 3d–f). The average length of the ZnO-NW was about 3.5 µm, and the diameter ranged from 10 to 40 nm. The AgNPs uniformly adhered on the ZnO-NW without aggregation, and the average diameter of AgNPs was 5.13 ± 1.26 nm (Figure 3c, n = 10).

ICP-MS analysis of the carbon cloth showed that the amount of Ag on AgNPs/ZnO-CC decreased significantly with time from 1.58 mg (0 day) to 0.50 mg (the fifth day) in MilliQ water treatment and 0.39 mg in the EPA medium treatment, respectively (Table 1). After five days, the loss of Ag from the carbon cloth was 68.4% and 75.3% in the MilliQ and EPA medium treatments, respectively. However,

by analyzing the treated water and Ag adsorbed to the surface of the tubes, only 0.215 mg Ag (13.6%) in the MilliQ medium and 0.063 mg Ag (4%) in the EPA medium could be recovered as release from AgNPs/ZnO-CC (Table 1). Additionally, for Zn, it was not possible to account for the mass loss of the cloths by analyzing the collected water (Table 1). The loss of Ag/Zn in the system may be due to fragmentation of the cloths followed by sedimentation of the bigger fragments that were not collected in the water samples for metal analysis. This is supported by the light microscopy and SEM analysis of the 0.45 μm filters which showed that all CC cloths released larger fragments during the release study (Figure S1).

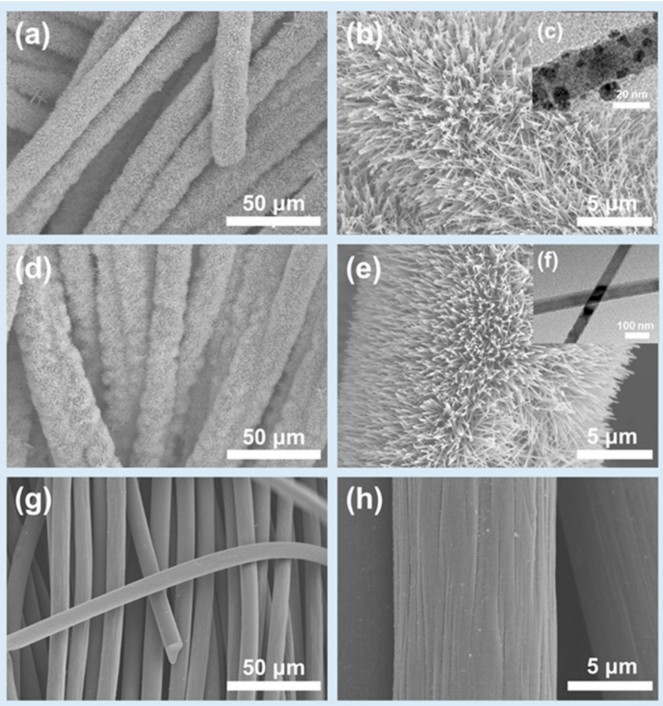

**Figure 3.** Characterization of the sterilization carbon cloth containing Ag/ZnO nanomaterials by transmission electron microscopy (TEM) and scanning electron microscope (SEM): (**a–c**) nanostructure of Ag/ZnO on surface of carbon cloth, (**a,b**) were acquired by SEM and (**c**) was acquired by TEM. (**d–f**) Nanostructure of ZnO growth on the surface of carbon cloth, (**d,e**) were acquired by SEM and (**f**) was acquired by TEM. (**g,h**) Structure of the carbon cloth, which was acquired by SEM.

**Table 1.** The mass balance of metals on the carbon cloth in the beginning and after five days.

| Metals | Zn (mg) | | | | | | Ag (mg) | |
|---|---|---|---|---|---|---|---|---|
| CC Type | CC | | ZnO-CC | | Ag/ZnO-CC | | Ag/ZnO-CC | |
| Medium | MilliQ | EPA | MilliQ | EPA | MilliQ | EPA | MilliQ | EPA |
| Remaining on CC (5 days) | 0.03 | 0.06 | 18.56 | 20.52 | 27.19 | 30.67 | 0.50 | 0.39 |
| Total release (medium) [1] | < 0.06 | < 0.06 | 1.90 | 0.12 | 1.54 | 0.28 | 0.20 | 0.06 |
| Adsorption on tubes | - | - | 0.04 | 0.98 | 0.15 | 0.25 | 0.015 | 0.003 |
| Recovery based on mass balance [2] | - | - | −3.2% | +3.2% | +11.8% | +20.5% | −55.8% | −71.6% |

[1] The total release of metals included measured dissolved and nanomaterials. Total release = sum of released masses day 1–5 (day 1 + day 2 + day 3 + day 4 + day 5). [2] The initial content of Zn (mg) on the carbon cloth was: 0.008 (CC), 19.82 (ZnO-CC), and 25.69 (Ag/ZnO-CC); the initial content of Ag (mg) on the carbon cloth was: 0.01 (CC), 0.0001 (ZnO-CC), and 1.58 (Ag/ZnO-CC). Recovery was based on mass balance = (initial content of metal — total release in medium — adsorption on tubes)/initial content × 100%.

### 3.2. Media Impact on Metal Releases from Sterilization Carbon Cloth Containing Ag/ZnO Nanomaterials

3.2.1. Silver Release

The total release of Ag from AgNPs/ZnO-CC in MilliQ water was 0.2 mg. In the parallel study using the EPA medium, a release of 0.06 mg was found (Table 1). The detected dissolved silver after ultracentrifugation at day 1 was lower than that by SP-ICP-MS in MilliQ water, but higher in the EPA medium. In the EPA medium, the two approaches showed different concentrations of dissolved silver on day 1 with 54.6 µg/L (ultracentrifugation) and 93.4 µg/L (SP-ICP-MS) (Figure 4f,h). The lower silver concentration in the EPA medium after ultracentrifugation could be due to the formation of insoluble Ag complexes during ultracentrifugation as shown by Miao et al. [36] and Ivask et al. [37]. For example, in artificial freshwater, only 33% of the added $AgNO_3$ was present in its ionic form, showing that 80% of Ag ions most likely settled during ultracentrifugation in the form of insoluble Ag species [37].

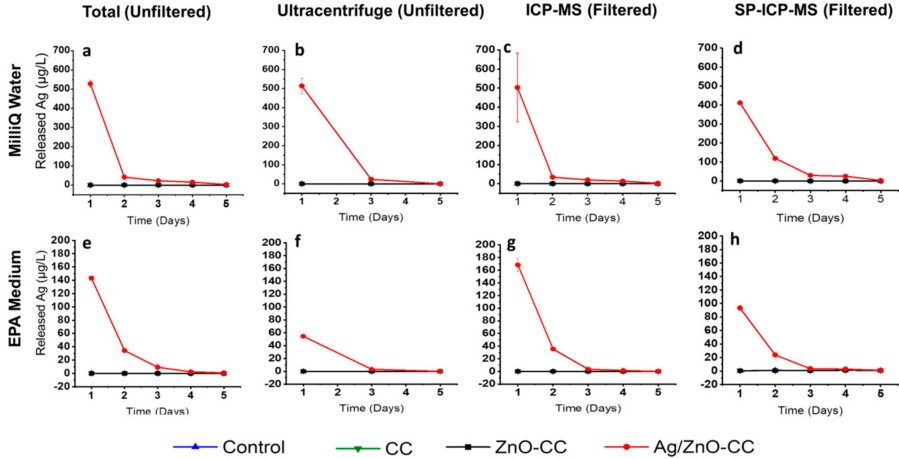

**Figure 4.** AgNPs/dissolved Ag release from carbon cloth in MilliQ and EPA media. (**a**,**e**) Total released AgNPs/dissolved Ag. (**b**,**f**) Dissolved Ag by ultracentrifugation analysis. (**c**,**g**) Released AgNPs/dissolved Ag analysis in filtered solution using ICP-MS. (**d**,**h**) Dissolved Ag analysis in filtered solution using SP-ICPMS. (Note: blue line = control, green line = CC, Black line = ZnO-CC are the background).

The analysis of AgNPs released by SP-ICP-MS showed very low concentrations (< 1.7 µg/L) in both the EPA medium and MilliQ water (Figures S2 and S3). This result suggests that only a limited aggregation of AgNPs (larger than 23 nm) occurred in the medium, because SP-ICP-MS is able to detect the size of nanoparticles larger than 23 nm in suspensions. However, adsorption of the released AgNPs on the inner surface of the tubes in the recirculation system was detected. The Ag analysis of acid washed tubes (the tubes themselves were not digested and analyzed) verified this loss in MilliQ water as 50.37 ± 2.69 µg/L Ag was found (Table 2). In experiments with the EPA medium, no Ag absorption on the surface of the tubes was found. However, in the EPA medium, we found that the amount of dissolved silver (supernatant after ultracentrifugation) was much lower than the total amount of silver. This indicates that there was a release of AgNPs and/or the formation of insoluble Ag complexes during ultracentrifugation in the EPA medium (Figure 4e,f). TEM analysis did not detect any AgNPs in either the EPA medium or in MilliQ water. The concentration of AgNPs from cloths in the EPA medium was estimated by Equation (1):

Total Ag release of 56.96 µg − Dissolved Ag in Filtered sample of 37.45 µg)/0.3 L = 65.02 µg/L    (1)

which is similar to the estimated concentration of released AgNPs in MilliQ water (50.37 µg/L). Thus, our results indicate that the medium can have an impact on the dissolution and speciation of released silver from the cloths, but not the release of AgNPs. Released AgNPs may play a role in the sterilization

efficiency of the system since other studies have shown that bacteria cells were inactivated due to cell membrane damage induced by AgNPs and graphene oxide nanosheets [38]. Furthermore, Loo et al. [39] reported that the fabrication of superabsorbent cryogels decorated with added AgNPs (PSA/AgNP cryogels) could increase water disinfection efficiency. This study revealed that both $Ag^+$ and $Ag^0$ are involved in the bactericidal mechanism of AgNPs. Significantly, bacterial cells exposed to PSA/$Ag^+$ cryogels did not show any cell–membrane damage even though the former had a higher cell-bound Ag concentration than that of the PSA/AgNP cryogels, thus indicating the differential action of $Ag^+$ and $Ag^0$ [39].

**Table 2.** Metals in washing water used to clean the experimental set up. Washing was carried out with MilliQ water and 10% $HNO_3$ after the termination of the 5-days' release. Average concentrations in washing water and standard deviations (n = 3).

| Washing Methods | Media | MilliQ Media | | EPA Media | |
|---|---|---|---|---|---|
| | Metals | Zn (µg/L) | Ag (µg/L) | Zn (µg/L) | Ag (µg/L) |
| MilliQ (6th day) | Control | < 20 | < 1 | < 20 | < 1 |
| | CC | < 20 | < 1 | < 20 | < 1 |
| | ZnO-CC | < 20 | < 1 | < 20 | < 1 |
| | Ag/ZnO-CC | < 20 | < 1 | 70.5 ± 4.16 | 1.11 ± 0.018 |
| 10% HNO3 (7th day) | Control | < 20 | < 1 | <1 | < 1 |
| | CC | < 20 | < 1 | 23.52 ± 3.78 | < 1 |
| | ZnO-CC | 142.52 ± 5.65 | < 1 | 327.3 ± 11.5 | < 1 |
| | Ag/ZnO-CC | 496.06 ± 120.6 | 50.37 ± 2.69 | 754.53 ± 18.75 | < 1 |

### 3.2.2. Zinc Release

During the five days of the recirculation experiment, the total amount of released zinc from the carbon cloth containing Ag/ZnO nanomaterials was 1.542 mg (6.0% of total Zn content) and 0.277 mg (1.1% of total Zn content) in MilliQ water and the EPA medium, respectively (Table 1). The presence of AgNPs on the surface of the ZnO-NW was found to have an impact on the zinc release. In the EPA medium, the released amount of zinc (0.277 mg) with AgNPs (ZnO/Ag-CC) was higher than the released amount (0.120 mg) without AgNPs (ZnO-CC). However, the opposite pattern was observed for the MilliQ medium (Table 1).

Samples collected at days 1, 3, and 5 were analyzed by SP-ICP-MS after filtration through a 0.45 µm membrane filter (Figure 5,d,f,h). In these samples, only dissolved Zn was detected, indicating no release of nano-sized fragments from the ZnO wires. The total Zn content in water samples was also analyzed, following ultracentrifugation and acid digestion. The amount of dissolved Zn detected by SP-ICP-MS was somewhat higher than that found after ultracentrifugation in both the EPA and MilliQ water (Figure 5b,d,f,h; Figure S6). This was to be expected since the samples analyzed by SP-ICP-MS were measured directly, thus minimizing the risk of losses during the handling of inhomogeneous samples.

In the collected water, the total Zn (Figure 5a,e) and dissolved Zn (ultracentrifuge sample, Figure 5b,f) were similar in the MilliQ water and EPA medium. After five days, the analysis of acid-washed tubes from the experimental system showed high Zn absorption on the surface of tubes in experiments with both EPA and MilliQ water (Table 2). The amount of absorbed Zn on tubes in experiments with the EPA media was twice as high as in those with MilliQ water. This indicates that the EPA medium showed a higher surface absorption of Zn, but a lower dissolution than that in MilliQ water. The release of dissolved Ag and Zn from the sterilization carbon cloth was higher in MilliQ water than in the EPA medium (Ag = 12.5% in MilliQ water and 3.6% in EPA medium; Zn 6.0% in MilliQ water and 1.1% in the EPA medium). This difference indicates that the release of Ag or Zn is not only impacted by the medium and size of the nanomaterials, but also by the experimental conditions (e.g., the concentration of the AgNPs, the static or flow-through experimental set up) and the embedding of nanomaterials in the product matrix [40,41].

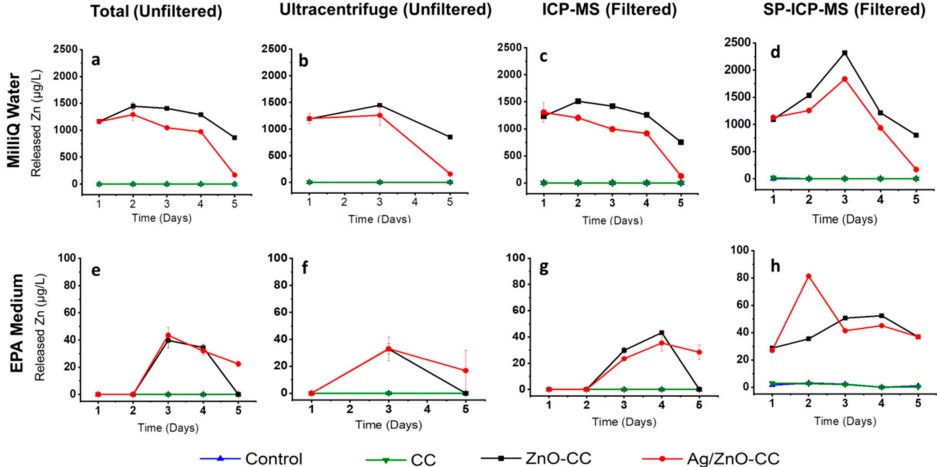

**Figure 5.** ZnO-NW/dissolved Zn released from the carbon cloth in MilliQ or EPA media. (**a**,**e**) Total release of ZnO-NW/dissolved Zn. (**b**,**f**) Dissolved Zn release measured by ultracentrifugation. (**c**,**g**) Released ZnO-NW/dissolved Zn in the filtered solutions using ICP-MS. (**d**,**h**) Dissolved Zn in the filtered solution measured by SP-ICPMS. (Note: the blue line of the control and green line of the CC are the background).

The releases of dissolved Ag and Zn were found to be time dependent following the same trend independent of the medium used. As shown in Figures 4 and 5, an initial high release was followed by a decrease in concentration to a lower and stable level. The highest Ag concentration in the experiments using MilliQ water was about 513 μg/L at day 1, which decreased to less than 23 μg/L at day 3, and further to less than 1 μg/L at day 5 (Figure 4b). The highest Ag concentration in experiments with the EPA medium was 54 μg/L at day 1, decreased to 3.1 μg/L after two days, and to less than 1 μg/L after five days (Figure 4f).

### 3.3. Indicative Risk Evaluation of Released Ag and Zn from Sterilization Carbon Cloth Containing Ag/ZnO Nanomaterials to Environment and Human Health

Both AgNPs and dissolved Ag are known to show high toxicity toward aquatic organisms [42,43]. Therefore, it is important to evaluate the potential risks associated with the release of AgNP or dissolved Ag to the aquatic environment and human health before using the new sterilization carbon cloth containing Ag/ZnO nanomaterials in real-world applications. In our study, the concentration of released AgNPs from the sterilization carbon cloth containing Ag/ZnO nanomaterials was 50 μg/L in both the EPA medium and MilliQ water, which was substantially higher than the AgNPs' predicted no effect concentration ($PNEC_{freshwater}$ = 0.012 μg /L) [44]. The concentration of released dissolved Ag was 143 μg/L in the EPA medium, which was also much higher than the PNEC estimate for ionic Ag ($PNEC_{freshwater}$ = 0.04 μg /L) [45].

In our study, the release of dissolved Zn in the EPA medium was 33 μg /L, and in addition, 755 μg/L of Zn accumulated on the surface of the tubes used in the experimental setup for the release simulation. The released concentrations of dissolved Zn were lower than the PNEC of Zn to freshwater organisms (Ionic Zn $PNEC_{Freshwater}$ = 69.2 μg /L) [46]. Therefore, it was concluded that only the silver released from the sterilization carbon cloth containing Ag/ZnO nanomaterials may pose risks to aquatic organisms (e.g., if applied as a sterilization step in wastewater treatment). However, this initial risk could be mitigated by rinsing the cloths with water prior to large-scale use and ensuring the safe disposal of the washing water. Using the quite low flow rates in this study a 3-day rinsing period was sufficient to reach a stable concentration level for the released metals. As shown in Figure 4, the release of dissolved Ag showed a sharp decrease in the first three days and then a slow decrease from the third day to the fifth day (dissolved Ag: MilliQ water = 2.6 ± 1.5 μg/L; EPA medium = 0.73 ± 0.63 μg/L). The minimum inhibitory concentrations (MIC) of AgNPs and Ag ions to *Escherichia coli* are 0.01 μg/L

and 0.02 µg/L [47], respectively, which indicates that the sterilization carbon cloth containing Ag/ZnO nanomaterials is still an effective antibacterial even after five days. However, the released silver may impact non-target organisms because the Ag concentration is higher than the PNEC in freshwater on the fifth day.

The toxicities of released metals to the non-target aquatic organisms call for a critical assessment of the potential environmental risks for real-world applications of the sterilization cloths. The chemical composition of the exposure medium (e.g., pH, ionic strength and composition, natural organic matters (NOM), temperature, and nanoparticle concentration) will interact to affect the aggregation or stabilization of nanomaterials containing Ag and Zn [26,40,48,49]. It remains to be studied how the composition of natural waters affects the efficiency of sterilization carbon cloth containing Ag/ZnO nanomaterials. The testing setup used in the present study does, however, present a relatively simple and cost-effective platform for realistic case-by-case evaluations.

The sterilization carbon cloth containing Ag/ZnO nanomaterials has the potential for not only being used in waste water treatment, but also in drinking water systems. A comparison of the dissolved Ag and Zn released from carbon cloth containing Ag/ZnO nanomaterials with the data from the World Health Organization (WHO) Guidelines for Drinking-Water Quality [49,50] shows that the release of dissolved Ag (143 µg/L) was slightly above the tolerated concentration of silver in drinking water (0.1 mg/L) [49]. However, there is no risk to humans if the sterilization carbon cloth is pre-washed in the two days prior as the dissolved Ag level dropped below the limit value (0.1 mg/L) after two days.

We calculated the exposure of dissolved Zn to humans through drinking water following the The European Chemicals Agency (ECHA) guidelines for oral exposure, assuming that an adult person (60 kg) drinks 2 L of water per day [34,51] and that all the water consumed is sterilized using the carbon cloth. The calculated exposure concentration of dissolved Zn to human by drinking water is as per Equation (2):

$$\text{Exposure to dissolved Zn: } 33 \text{ µg/L} \times 2 \text{ L (water) per day}/60 \text{ kg} = 1.10 \text{ µg /kg bw/day} \tag{2}$$

Our calculation showed that the potential exposure of dissolved Zn from the new sterilization carbon cloth (1.10 µg /kg bw/day) was much lower than a provisional maximum tolerable daily intake (PMTDI) of 1.0 mg/kg of body weight [51].

We also evaluated the potential risks of released nano-scaled particles from the sterilization carbon cloth to human health. We did not detect any ZnO-NW in the water phase and hence no risk to human health is anticipated. However, the released AgNPs (50 µg/L) from the sterilization carbon cloth might pose risks to human health. To estimate the order of magnitude of this risk, we applied the margin of exposure (MoE) approach as described in our previous study to evaluate the potential risk to human health posed by AgNPs in drinking water [25]. We calculated the MoE and reference MoE for our study. If the MoE > reference MoE, the exposure level is not high enough to be of concern and mitigation may not be required. A reference MoE was estimated, borrowing from the approach previously published by the U.S. EPA (Equation (3)) [52].

$$\text{Reference MoE long-term} = 10 \text{ (AF}_{inter}) \times 10 \text{ (AF}_{intra}) \times 10 \text{ (UFD)} \times 3 = 3000 \tag{3}$$

Based on the toxicity data (PoD: 30 mg/kg/day) from Kim et al. (2008) and the exposure AgNPs data from our study (Equation (4)):

$$\text{AgNPs: } 50 \text{ µg/L} \times 2 \text{ L (water) per day}/60 \text{ kg} = 1.67 \text{ µg/kg bw/day} \tag{4}$$

We estimated a MoE of human to AgNPs (Equation (5)):

$$\text{MoE} = \text{PoD/Daily Dose} = 30{,}000 \text{ µg/kg/day}/1.67 \text{ µg/kg/day} = 18{,}000 \tag{5}$$

The results showed that the value of the MoE (18,000) was much higher than the value of the reference MoE (3000). In conclusion, the estimations indicate no human health risks associated with exposure to AgNPs with the application of the carbon cloths to a drinking water sterilization system.

## 4. Conclusions

Quantitative analysis of the release of nanomaterials from nano-enabled consumer products is urgently required to conduct realistic risk assessments of nanomaterials to human health and the environment. This study evaluated the release of Ag and Zn from sterilization carbon cloth containing Ag/ZnO nanomaterials, a promising technology in water treatment. It was found that the aquatic media had a significant impact on the nanoparticle release and dissolution in an aquatic environment. The quantitative analysis of the released Ag and Zn in MilliQ water and EPA medium indicated a potential risk associated with both metals to the non-target aquatic organisms and thus the real-life applications of this technology in an aquatic environment should be carefully evaluated. However, the assessment of human health risks from the sterilization carbon cloths showed no risks if the sterilization carbon cloth was pre-washed two days prior, and thus application of this technology was deemed to be safe in a drinking water system. Furthermore, it was found that a SP-ICP-MS approach provides better characterization of the dissolution of nanomaterials when compared to the analysis of dissolved metal after ultracentrifugation. The approach presented in this study to assess the release of metals from sterilization cloths allows for simple, but cost-effective, case specific evaluations that may be crucial for environmental risk assessments accounting for the natural water composition. Thus, this study provides an exposure and risk methodology that could be used to evaluate nano-enabled solutions used for water treatment.

**Supplementary Materials:** The following are available online at http://www.mdpi.com/2073-4441/11/11/2276/s1, Figure S1: Image analysis of filtered fragments by the light microscopy, Figure S2: Released Ag nanoparticles from carbon clothes, Figure S3: Size distribution of released Ag nanoparticles from carbon cloth in 5 days' simulation of release, Figure S4: Compare different analysis methods to analyse the released ZnO-NW/dissolved Zn and/or AgNP/dissolved Ag from the carbon cloth in a 5 day simulation, Figure S5: The output of statistical analysis to released metals.

**Author Contributions:** C.P., J.T., and A.M. carried out the experiment. C.P. wrote the manuscript. A.B., Z.L. and H.F. review and edit the manuscript.

**Funding:** The research was funded by the European Union's Seventh Framework Program and the Horizon 2020 Research and Innovation Program under Marie Sklodowska-Curie Actions (grant no. 713683 (H2020)), the H.C. Ørsted Postdoc Program, co-funded by Marie Skłodowska-Curie Actions, and was funded partially by the National Natural Science Foundation of China (No. 61875015, 31571006, 81601629) and the Beijing Natural Science Foundation (2182091).

**Conflicts of Interest:** The authors declare no conflicts of interest.

## Abbreviations

| | |
|---|---|
| TENG | Wave-driven triboelectric nanogenerator |
| CFU | Colony forming units |
| ZnO-NW | ZnO-nanowires |
| AgNPs | Silver nanoparticles |
| EF | Electrical field |
| ROS | Reactive oxygen species |
| SEM | Scanning Electron Microscope |
| TEM | Transmission Electron Microscopy |
| EDX | Energy Dispersive X-Ray Analysis (EDX) |
| ICP-MS | Inductively Coupled Plasma Mass Spectrometry |
| SP-ICP-MS | Single Particle Inductively Coupled Plasma Mass Spectrometry |

CC	Carbon cloth (CC)
ZnO-CC	Carbon cloth containing ZnO-NW
Ag/ZnO-CC	Carbon cloth containing AgNPs and ZnO-NW
NOM	Natural organic matters

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
