# Peer review of "Release of Ag/ZnO Nanomaterials and Associated Risks of a Novel Water Sterilization Technology"

_water, doi:10.3390/w11112276_

Round 1

Reviewer 1 Report

The authors were investigated the risk of Ag and ZnO leaching during the water treatment by carbon cloths. The authors were designed the work systematic way with performing some valuable experimental works. It is also necessary to critically evaluate new data and do not make hasty conclusions which may lead to misinterpretations. However, several points are important to be addressed before going to possible publication in this journal. Also, the authors need to address all points in the revision stage for broad range readers understanding.

-Abstract: This section is completely different than the Introduction and Experimental sections. The main findings with important opinions are acceptable. The mathematical terms need to be added. The authors need to consider these points in the revision stage.

-The English language need to check carefully in the revision stage because of careless mistake in several positions. In addition, the manuscript should be thoroughly checked for English corrections as there are some colloquial terms being used.

-References: There are many references are not adjacent with this study. The authors need to take note in the revision stage and cite relevant references including high impact journal to make the manuscript in broad range readers. In addition, there are many references are not up to dated and used the last 10 years or latest. Also the numbers of references need to be extended to comply the novelty and discussion.

-There are many studies reported in the literature regarding the water purification based on the ligand based composite materials. Based on this, do the authors think that the present study is an improvement when compared to other materials? If so, please provide some discussions on the advantage and disadvantage as the novelty in the Introduction section. And the authors should highlight the scientific value of the present work with citing high impact journals such as Chemical Engineering Journal, 289 (2016) 65–73; Chemical Engineering Journal, 266 (2015) 368–375; Chemical Engineering Journal, 236 (2014) 100–109; Chemical Engineering Journal, 273 (2015) 286–295; Chemical Engineering Journal, 343 (2018) 118–127; Chemical Engineering Journal, 334 (2018) 432–443; Chemical Engineering Journal, 332 (2018) 377–386; Chemical Engineering Journal, 228 (2013) 327–335; Chemical Engineering Journal, 363 (2019) 64–72; Chemical Engineering Journal, 331 (2018) 54–63; Sensors and Actuators B: Chemical, 206 (2015) 692–700; Chemical Engineering Journal, 259 (2015) 611–619; Chemical Engineering Journal, 279 (2015) 639–647; Materials Science and Engineering: C, 101 (2019) 686–695; Journal of Molecular Liquids, 284 (2019) 502–510. The authors need to take note of the above references in the revised manuscript.

-As a complete scientific paper, the manuscript makes no mention of the preparation of the conjugate material from beginning to end, which is puzzling and unprofessional.

--The optimum condition need to be determined. The authors need to pay attention in the revision.

-Result and discussion section (including materials characterization) must explain with more experimental findings, such as experimental results, significance of work, finding of results and choice of materials. Result and discussion part must be supported in the main manuscript by the following references: Microporous and Mesoporous Materials, 196 (2014) 261–269; Journal of Cleaner Production, 228 (2019) 1311–1319; Journal of Environmental Chemical Engineering, 7 (2019) 103124; Composites Part B: Engineering, 172 (2019) 387–396; Journal of Environmental Chemical Engineering, 7 (2019) 103087; Journal of Environmental Chemical Engineering,7 (2019) 103378.

-In the results and discussions part, the authors only presented the experimental results simply. More detailed mechanism analyses are needed to explain why the present material is excellent and how it works.

I WOULD LIKE TO SEE THE REVISED MANUSCRIPT

Author Response

Comments and Suggestions for Authors

The authors were investigated the risk of Ag and ZnO leaching during the water treatment by carbon cloths. The authors were designed the work systematic way with performing some valuable experimental works. It is also necessary to critically evaluate new data and do not make hasty conclusions which may lead to misinterpretations. However, several points are important to be addressed before going to possible publication in this journal. Also, the authors need to address all points in the revision stage for broad range readers understanding.

Response: Many thanks to the reviewer for the suggestions to improve the manuscript. We have addressed the points which were mentioned by reviewer below and in the revised manuscript.  

-Abstract: This section is completely different than the Introduction and Experimental sections. The main findings with important opinions are acceptable. The mathematical terms need to be added. The authors need to consider these points in the revision stage.

Response: We are not quite sure what the reviewer hint at here. In our view an abstract should be different than Introduction/Experimental section but should of course cover the content of the paper accurately. We have carefully reviewed and made changes to the abstract to ensure complimentary between abstract and the manuscript. In doing so, we found it difficult to add mathematical term, still keep the focus and the word limitation. Hence we hope our changes are considered acceptable.

-The English language need to check carefully in the revision stage because of careless mistake in several positions. In addition, the manuscript should be thoroughly checked for English corrections as there are some colloquial terms being used.

Response: We thank the reviewer for this observation and have carefully edited the paper with this focus in mind.

-References: There are many references are not adjacent with this study. The authors need to take note in the revision stage and cite relevant references including high impact journal to make the manuscript in broad range readers. In addition, there are many references are not up to dated and used the last 10 years or latest. Also the numbers of references need to be extended to comply the novelty and discussion.

Response: The references are up to dated and included in discussion (Page 10).

-There are many studies reported in the literature regarding the water purification based on the ligand based composite materials. Based on this, do the authors think that the present study is an improvement when compared to other materials? If so, please provide some discussions on the advantage and disadvantage as the novelty in the Introduction section. And the authors should highlight the scientific value of the present work with citing high impact journals such as Chemical Engineering Journal, 289 (2016) 65–73; Chemical Engineering Journal, 266 (2015) 368–375;

Chemical Engineering Journal, 236 (2014) 100–109;

Chemical Engineering Journal, 273 (2015) 286–295;

Chemical Engineering Journal, 343 (2018) 118–127;

Chemical Engineering Journal, 334 (2018) 432–443;

Chemical Engineering Journal, 332 (2018) 377–386;

Chemical Engineering Journal, 228 (2013) 327–335;

Chemical Engineering Journal, 363 (2019) 64–72;

Chemical Engineering Journal, 331 (2018) 54–63;

Sensors and Actuators B: Chemical, 206 (2015) 692–700;

Chemical Engineering Journal, 259 (2015) 611–619;

Chemical Engineering Journal, 279 (2015) 639–647;

Materials Science and Engineering: C, 101 (2019) 686–695;

Journal of Molecular Liquids, 284 (2019) 502–510.

The authors need to take note of the above references in the revised manuscript.

Response: Very appreciated the reviewer to provide the detailed list for references. The references are added in introduction (Page 2).

The advantage of our method is to enhance the silver nanoparticles capacity of removing pathogenic bacteria from water.  All the papers mentioned were considered in our revision of the manuscript and some of them were added and used in the discussion section of the revised manuscript. It is important to note that the methods described in the papers suggested by reviewer #1 are aimed at removal of organic or inorganic materials from water. This is not comparable to our method which is used to remove bacteria from water.

From rev #1's comments we have realized that there may have been a confusion regarding the purpose of the water treatment methodology we are evaluated. To avoid any confusion, we change the title of our paper from " Evaluation of water purification technology: Release of Ag/ZnO nanomaterials and associated risks " to “Release of Ag/ZnO nanomaterials and associated risks of a novel water sterilization technology”.

-As a complete scientific paper, the manuscript makes no mention of the preparation of the conjugate material from beginning to end, which is puzzling and unprofessional.

Response: The method of preparation of the materials was described in section 2.1. To keep our paper short we added the citation of the method from our previous study. We have now expanded this to accommodate the reviewer’s relevant comment. In the revised version section 2.1 now reads like this:

2.1 Synthesis and characterization of sterilisation Ag/ZnO nanomaterials

The synthesis of sterilization Ag/ZnO nanomaterials was followed our previous study (Tian et al., 2017).A woven textile material made of carbon fiber (10 μm in diameter) was used as the substrate. The carbon cloth was cleaned with acetone and ethanol three times firstly, and then treated by oxygen plasma for 5 min before nanomaterial growth. After plasma treatment, ZnO-NW seeds solution was added to carbon cloth to initiate the ZnO nanowire growth. The ZnO nanowires were grown in aqueous solution at 95 ℃ for 4 h by a wet-chemical method. After that, the ZnO nanowires samples were immersed in 100 mM AgNO3 ethanol-water solution for 24 h in darkness, then rinsed with ethanol to remove unconjugated Ag ions. At last, both sides of the composites sample were exposed under UV (370 nm) irradiation for 0.5 h to reduce the Ag ions to Ag NPs in situ on ZnO nanowire. (page 6).

The optimum condition need to be determined. The authors need to pay attention in the revision.

Response: We are not sure which optimum condition the reviewer refers to and were therefore not able to take this comment into account in our revision.

-Result and discussion section (including materials characterization) must explain with more experimental findings, such as experimental results, significance of work, finding of results and choice of materials. Result and discussion part must be supported in the main manuscript by the following references: Microporous and Mesoporous Materials, 196 (2014) 261–269; Journal of Cleaner Production, 228 (2019) 1311–1319; Journal of Environmental Chemical Engineering, 7 (2019) 103124; Composites Part B: Engineering, 172 (2019) 387–396; Journal of Environmental Chemical Engineering, 7 (2019) 103087; Journal of Environmental Chemical Engineering,7 (2019) 103378.

Response: We improved the results and discussion sections. We are sorry that the references provided by the reviewer may be not helpful to support our study since these references are all focus on the inorganic or organic materials remove from water. Our study focuses on pathogenic bacteria remove. We added papers on AgNPs and removal of bacteria to our discussion in the revision (Page 10).

-In the results and discussions part, the authors only presented the experimental results simply. More detailed mechanism analyses are needed to explain why the present material is excellent and how it works.

Response: many thanks for the comments. We added the mechanisms’ description in our revision (Page 10).

Reviewer 2 Report

The papers reports the release of Ag and Zn nanomaterials from carbon cloth in water. The release of nanoparticles is not a new topic however highly relevant. For this reason, some points should be considered before publication.

1- The authors should provide information regarding the calibration curves used in the ICP-MS analysis. This should be added to the experimental section.

2- In section 3.2.2., the zinc samples have been diluted by a factor of 1000. How can the dilution cause any extra uncertainty?  Have the authors considered Zn and Ag mass transport within the reactor? Which other factors could contribute to the uncertainties?

3- Zn and Ag diffusion coefficients in water and solubility product constants (Ksp) should be considered when drawing conclusions of the studied system.

Author Response

The papers reports the release of Ag and Zn nanomaterials from carbon cloth in water. The release of nanoparticles is not a new topic however highly relevant. For this reason, some points should be considered before publication.

Response: the three points were addressed in our revision.

The authors should provide information regarding the calibration curves used in the ICP-MS analysis. This should be added to the experimental section.

Response: The calibration curves was added to the experimental section.

2- In section 3.2.2., the zinc samples have been diluted by a factor of 1000. How can the dilution cause any extra uncertainty?  Have the authors considered Zn and Ag mass transport within the reactor? Which other factors could contribute to the uncertainties?

Response: The reviewer’s comment made us aware that this statement was not precise enough. The fact that on unit operations or treatments are involved in the SP-ICP-MS analyses result in a lower loss. We have stated this in the revised manuscript.

3- Zn and Ag diffusion coefficients in water and solubility product constants (Ksp) should be considered when drawing conclusions of the studied system.

Response: We thank the reviewer for this observation and agree in principle. However, Ksp (as well as diffusion coefficients) would only related to the dissolved metals and not to the nanomaterials (for which such constants do not exist). For these the rate constants for dissolution would be of higher interest – but these are unfortunately also yet a topic of scientific study and presently not possible to generalize.

Reviewer 3 Report

Dear Authors, you can improve this article by adding more comparison research with other countries experience.

Author Response

Dear Authors, you can improve this article by adding more comparison research with other countries experience.

Response: the comparison with a strengthened international outlook was added in the revision.

Round 2

Reviewer 1 Report

The authors were considered all remarks of the reviewers and revised the manuscript accordingly. This revised manuscript can be accepted.